Characterization and complete genome sequence of Privateer, a highly prolate Proteus mirabilis podophage

Corban James E. 1 2 3
http://orcid.org/0000-0002-3774-5896 Ramsey Jolene 1 2 jolenerr@tamu.edu
1 Department of Biochemistry & Biophysics, Texas A&M University , College Station, TX , USA
2 Center for Phage Technology, Texas A&M University , College Station, TX , USA
3 Department of Biochemistry, University of Wisconsin-Madison , Madison, WI , USA
Breitbart Mya
Electronic publication date: 2021 Feb 10
Publication date: 2021
Volume: 9
Electronic Location ID: e10645
Received 2020 Oct 20; Accepted 2020 Dec 3
Copyright: © 2021 Corban and Ramsey
Copyright year: 2021
Copyright holder: Corban and Ramsey
License: This is an open access article distributed under the terms of the Creative Commons Attribution License, which permits unrestricted use, distribution, reproduction and adaptation in any medium and for any purpose provided that it is properly attributed. For attribution, the original author(s), title, publication source (PeerJ) and either DOI or URL of the article must be cited.
License URL: https://creativecommons.org/licenses/by/4.0/

Keywords: Bacteriophage, Genomics, Urinary tract infection, Proteus, Prolate

Funding: National Science Foundation DBI-1565146 Center for Phage Technology (CPT) Texas A&M University and Texas AgriLife Department of Biochemistry and Biophysics NIH 1S10RR025111-01 This work was supported by funding from the National Science Foundation (award DBI-1565146). Additional support came from the Center for Phage Technology (CPT), an Initial University Multidisciplinary Research Initiative supported by Texas A&M University and Texas AgriLife, and from the Department of Biochemistry and Biophysics. Funds for the purchase of the Orbitrap Velos Pro mass spectrometer in the Institutional Mass Spectrometry Laboratory of the University of Texas Health Science Center at San Antonio were provided by NIH grant 1S10RR025111-01 to Susan T. Weintraub. Susan T. Weintraub directed the mass spectronomy analysis service. The funders had no role in study design, data collection and analysis, decision to publish, or preparation of the manuscript.

==============================
The Gram-negative bacterium Proteus mirabilis causes a large proportion of catheter-associated urinary tract infections, which are among the world’s most common nosocomial infections. Here, we characterize P. mirabilis bacteriophage Privateer, a prolate podophage of the C3 morphotype isolated from Texas wastewater treatment plant activated sludge. Basic characterization assays demonstrated Privateer has a latent period of ~40 min and average burst size around 140. In the 90.7 kb Privateer genome, 43 functions were assigned for the 144 predicted protein-coding genes. Genes encoding DNA replication proteins, DNA modification proteins, four tRNAs, lysis proteins, and structural proteins were identified. Cesium-gradient purified Privateer particles analyzed via LC-MS/MS verified the presence of several predicted structural proteins, including a longer, minor capsid protein apparently produced by translational frameshift. Comparative analysis demonstrated Privateer shares 83% nucleotide similarity with Cronobacter phage vB_CsaP_009, but low nucleotide similarity with other known phages. Predicted structural proteins in Privateer appear to have evolutionary relationships with other prolate podophages, in particular the Kuraviruses

Introduction

Proteus mirabilis, a ubiquitous Gram-negative bacterium, is most frequently isolated from the gastrointestinal tracts of humans and animals (O’Hara, Brenner & Miller, 2000; Drzewiecka, 2016). Human infections with P. mirabilis often occur in the eyes, mouth, and intestines, yet are predominantly associated with urinary tract infections (UTIs) (Schaffer & Pearson, 2015; Armbruster, Mobley & Pearson, 2018). P. mirabilis causes between 10% and 44% of catheter-associated urinary tract infections (CAUTIs), which are among the world’s most abundant nosocomial infections. CAUTIs typically occur in elderly patients who undergo long-term catheterization (Schaffer & Pearson, 2015; Milo et al., 2017). Additionally, high-mortality (up to 50% in elderly patients) bacteremia and sepsis cases associated with P. mirabilis most frequently occur following UTIs and CAUTIs. Increasing rates of antimicrobial resistance in clinical P. mirabilis strains is a grave concern for nosocomial P. mirabilis infection severity (Luzzaro et al., 2001; Wang et al., 2014; Girlich et al., 2020). Alternative treatment avenues for Proteus-based infections are urgently needed (Milo et al., 2017; Alves et al., 2019).

Phages have been investigated as a treatment method for Proteus-based CAUTIs and associated catheter-blockages, for which established control strategies are lacking (Milo et al., 2017; Maszewska et al., 2018; Ujmajuridze et al., 2018; Gomaa et al., 2019). Byproducts of P. mirabilis biofilms in CAUTI cases lead to the production of crystalline aggregates which, in combination with biofilm growth, can dangerously obstruct the flow of a catheterized individual’s urine (Schaffer & Pearson, 2015; Milo et al., 2017). Multiple studies and patient trials have demonstrated the capacity for phages to inhibit the formation of P. mirabilis biofilms on catheter surfaces, which merits the further investigation and classification of P. mirabilis phages (Milo et al., 2017; Maszewska et al., 2018; Ujmajuridze et al., 2018; Gomaa et al., 2019). As of writing (August 2020), fewer than 30 Proteus phage genome sequences are deposited in the public NCBI Genbank database (Sayers et al., 2020). The few characterized Proteus-specific phages are widely variable in morphology, genome sequence length, host range, and gene content (Prozesky, Klerk & Coetzee, 1965; Alves et al., 2019).

In this study, classic growth assays and particle characterization demonstrate Privateer is a highly prolate (or elongated) podophage belonging to the rare C3 morphotype (Ackermann & Eisenstark, 1974; Ackermann, 2001), and genomics suggest it is most closely related to the Kuraviruses.

Materials and Methods

Phage isolation

Bacteriophage Privateer was isolated from filtered (0.2 μm pore size) activated sludge sourced from a wastewater treatment facility in Navasota, TX. The phage was propagated on P. mirabilis strain ATCC 35659 aerobically at 30 °C in Brain Heart Infusion (BHI, BD) media using the previously described soft-agar overlay method (Adams, 1956). Individual plaques that developed overnight on host lawns were picked into media and sterilized with CHCl3. The plaque purification was repeated three times prior to additional experiments.

To prepare high-titer lysates, a culture of ATCC 35659 was grown at 30 °C with 180 rpm shaking in BHI broth to OD550 of 0.2 (~5 × 107 cfu/mL). Phage were then adsorbed to the cells at MOI = 0.1. The cell+phage mixture was incubated for two hours (30 °C with aeration at 180 rpm) to achieve culture lysis. The phage lysate was sterilized with CHCl3. Privateer was stored at 4 °C.

Phage purification

Privateer stocks were further concentrated and purified as previously described (Zeng et al., 2010). First, 500 mL of phage lysate was treated with 1 μg/mL of DNase/RNase (VWR), followed by precipitation with 10% wt/vol PEG-8000 and 1 M NaCl. The particles were resuspended in SM buffer (100 mM NaCl, 1 M MgSO4, 1 M Tris-HCl pH 7.5). Next, CHCl3 was added 1:1 to the suspension to separate the PEG-8000 from the phages. This concentrated phage lysate was ultracentrifuged using a CsCl step gradient (1.3 g/mL, 1.5 g/mL, and 1.7 g/mL CsCl), extracted, and subjected to CsCl equilibrium centrifugation. The band corresponding to phage Privateer was extracted (to obtain a volume of ~400-μL) and subsequently dialyzed three times against 600 mL of SM buffer.

Phage morphology determination

CsCl-purified Privateer particles were negatively stained with 2% uranyl acetate via the Valentine method and viewed by transmission electron microscopy at the Texas A&M Microscopy and Imaging Center (Valentine, Shapire & Stadtman, 1968).

Adsorption assay

Brain Heart Infusion was inoculated 1:200 with an overnight culture of P. mirabilis ATCC 35659, then incubated shaking at 30 °C until reaching exponential phase at OD550 0.2. After chilling on ice for 10 min, the cells were pelleted (10,000×g, 10 minutes, 4 °C), then resuspended in cold BHI. Phage were added at an MOI = 0.01 and incubated on ice. At each time point, a 200 μL aliquot was taken from the cell+phage mixture and centrifuged (17,000×g, 2 min, 4 °C). The residual phage particles in the aliquot supernatants were immediately quantified by soft-agar overlay plaque assay. The fraction of adsorbed phages at each time point was calculated as the titer of the supernatant divided by the titer at the initial time point (Hernandez-Morales et al., 2018).

One-step growth curve

The latent period and burst size of Privateer were assayed via a one-step growth curve experiment performed in triplicate as in Hernandez-Morales et al. (2018). P. mirabilis ATCC 35659 was grown to exponential phase and Privateer phage was adsorbed at MOI = 0.01 as described above. After a 20 min cold incubation, cells were pelleted (10,000×g, 5 min, 4 °C) to remove non-adsorbed phages and resuspended in fresh, cold BHI. The cells+phage mixture was then diluted into 1:2,000 into pre-warmed BHI and incubated at 30 °C with 225 rpm shaking. At each time point, a sample was plated in Nutrient Broth (BD Difco) 0.75% top agar with P. mirabilis ATCC 35659 on Nutrient Broth 1.5% agar plates. The zero time point was also treated with chloroform prior to plating to allow measurement of infective centers. Burst size in pfu per infected cell was calculated as the ratio of ending phage titer to the number of infective centers (starting titer minus free phage titer as measured with chloroform treatment). Average and standard deviation were calculated from three replicates.

Lysis assays

P. mirabilis ATCC 35659 was grown and infected with Privateer at MOI = 10 as described above. OD550 measurements of the culture were taken over time.

From MOI = 3 infections, 1.5 ul samples were observed between 60 and 80 min post-infection on a glass slide with a coverslip at 100×g magnification under oil immersion on a Zeiss Axio Observer seven inverted microscope through lysis of single cells. Time series images were processed using the Zeiss Zen 2.3 software.

Host range

The host range of Privateer was analyzed by pipetting 10 μL quantities of serial dilutions of a high titer lysate (~1010 pfu/mL) on lawns of selected Proteus mirabilis strains. All strains, except B446 which was acquired from GangaGen Biotechnologies Private Limited (India), were from ATCC (Table 1). Expanded host range Gammaproteobacteria strains were Escherichia coli MG1655 (lab stocks), Klebsiella pneumoniae 1776c, a pKpQIL plasmid-cured derivative of K. pneumoniae strain 1776c (Satlin et al., 2017), and Serratia marcescens D1 (no. 8887172; Ward’s Science). Bacteria were considered sensitive to Privateer if single plaques were observed at phage concentrations above the detection limit established with the isolation host (≥102 pfu/mL) after incubation at 30 °C. Similar results were observed in three replicates.

Table 1 Privateer host range among P. mirabilis strains and other Gammaproteobacteria.

Bacterial host	Sensitive to Privateer	Description	
P. mirabilis ATCC 35659	+	Single plaques at ≥102 pfu/mL	
P. mirabilis B446	–	No clearing	
P. mirabilis ATCC 7002	+	Single smaller plaques at ≥106 pfu/mL	
P. mirabilis ATCC 29906	+	Single smaller plaques at ≥106 pfu/mL	
P. mirabilis ATCC 43701	–	Faint clearing at ≥109 pfu/mL	
P. mirabilis ATCC BAA-856	+	Single smaller plaques at ≥106 pfu/mL	
P. mirabilis ATCC 25933	+	Single smaller plaques at ≥106 pfu/mL	
E. coli MG1655	–	No clearing	
K. pneumoniae 1776c	–	No clearing	
S. marcescens D1	–	No clearing	
Note:

Proteus strains were considered sensitive if single plaques were observed at higher dilutions of the 1010 pfu/mL phage lysate. The bolded strain (ATCC 35659) indicates the host used for Privateer isolation.

gDNA preparation, sequencing, and assembly

Genomic DNA was purified from the phage as previously described with the Promega Wizard DNA clean-up system (Summer, 2009) after PEG precipitation, prepared as Illumina TruSeq Nano low-throughput libraries with 550-bp inserts using a Nextera DNA Flex Library Prep kit, and sequenced in paired-end 250-bp reads via Illumina MiSeq v2 300-cycle chemistry. The 396,576 sequence reads from the index containing the phage genome were quality controlled with FastQC (http://www.bioinformatics.babraham.ac.uk/projects/fastqc/). The phage genome was assembled into a single raw contig via SPAdes v.3.5.0 with 636.7-fold coverage after trimming with the FASTX-Toolkit 0.0.14 (http://hannonlab.cshl.edu/fastx_toolkit/) (Bankevich et al., 2012). PCR amplification across the raw contig ends (forward primer 5′-ctcgttaccagcgcagaaa-3′ and reverse primer 5′-caggtgctaaccaaggtttagg-3′) accompanied by Sanger sequencing of the DNA product verified that the contig sequence was complete. Analyses with PhageTerm indicate a novel ends type (Garneau et al., 2017).

Genome annotation

The Privateer complete contig was assembled, analyzed, and annotated in the Center for Phage Technology Galaxy and Web Apollo interfaces (https://cpt.tamu.edu/galaxy-pub), as described by Ramsey et al. (2020). Briefly, protein-coding gene predictions relied on Glimmer v3.0 and MetaGeneAnnotator v1.0 (Delcher et al., 2007; Noguchi, Taniguchi & Itoh, 2008). ARAGORN v2.36 was run to detect tRNA coding sequences (Laslett & Canback, 2004). Seven rho-independent termination sites were annotated using TransTermHP v2.09 when scores were >90, the stem was at least 5 bp in length, and at least four T’s were downstream (Kingsford, Ayanbule & Salzberg, 2007). For functional assignments, predicted gene functions were assigned using InterProScan v5.33-72, , BLAST v2.2.31 with a 0.001 maximum expectation value, and TMHMM v2.0 at the default settings (Krogh et al., 2001; Camacho et al., 2009; Jones et al., 2014). All BLAST queries were run against the NCBI nonredundant and UniProtKB Swiss-Prot and TrEMBL databases (Coordinators et al., 2017; Consortium, 2019).

Supporting analysis was performed using the HHSuite v3.0 HHpred tool (multiple sequence alignment generation with the HHblits ummiclus30_2018_08 database and modeling with PDB_mmCIF70) (Zimmermann et al., 2018).

The genome sequence and associated data for phage Privateer were deposited under GenBank accession no. MT028297, BioProject accession no. PRJNA222858, Sequence Read Archive accession no. SRR11024936 and BioSample accession no. SAMN14002513. Proteus phage Privateer is the proposed species submitted to the International Committtee on Taxonomy of Viruses (ICTV) for consideration. The genome map was prepared using the Genome Linear Plot tool in CPT Galaxy, modified from DNA features viewer (Zulkower & Rosser, 2020; Ramsey et al., 2020).

Comparative genomics

Initial Privateer DNA sequence similarity to other phage genomes was determined using the progressiveMauve v2.4.0 alignment algorithm coupled with MIST (Darling, Mau & Perna, 2010; Ramsey et al., 2020). Protein similarity with other phages was calculated with BLAST v2.2.31 using a 0.001 maximum expectation value and the relatedness tools in CPT Galaxy.

Records for the top related phage classifications, including many unclassified podoviruses and several in the genus Kuravirus as presented in the NCBI Taxonomy and the International Committee on Viral Taxonomy (Lefkowitz et al., 2017; Schoch et al., 2020), were retrieved for comparison (Table 2). C3 podophage proteins were used to generate a BLAST database and each protein was queried against this database via BLASTp (expectation value cutoff = 10−20). A greedy algorithm grouped the coding sequences into clusters, which were aligned with ClustalW and presented as locally colinear blocks (LCBs) in an XMFA file (Larkin et al., 2007). Adjacent LCBs below the threshold distance of 50 nucleotides were consolidated. X-Vis, a custom XMFA visualization tool created in JavaScript, rendered the genomes and their LCB alignments (Ramsey et al., 2020). Figures were sized and labelled in Inkscape (Inkscape Project, 2020).

Table 2 Comparative genomics phage accessions.

Host	Podophage	Genbank accession	
Kuravirus Phages	
Escherichia coli	phiEco32	EU330206	
Escherichia coli	Paul	MN045231	
Escherichia coli	vB_EcoP_SU10	KM044272	
Escherichia coli	KBNP1711	KF981730	
Escherichia coli	LAMP	MG673519	
Escherichia coli	EP335	MG748548	
Escherichia coli	ECBP2	JX415536	
Non-Kuravirus Phages	
Salmonella enterica	7-11	HM997019	
Salmonella enterica	SE131	MG873442	
Cronobacter sakazakii	vB_CsaP_GAP52	JN882286	
Cronobacter sakazakii	vB_CsaP_009	LC519601	
Aeromonas hydrophila	LAh_6	MK838112	
Vibrio parahaemolyticus	Vp_R1	MG603697	
Serratia marcescens	KSP100	AB452992*	
Note:

* Only contains structural genes (orf1 to orf8).

The hosts and GenBank accessions of the C3 podophages used for comparative genomics in this report.

Analysis of phage structural proteins by mass spectrometry

All phage protein analyses were performed using samples from the CsCl-purified Privateer stock. The total protein mass in the CsCl-purified stock was determined via Bradford assay according to manufacturer recommendations (Bio-Rad). To get a view of structural protein complexity prior to mass spectrometry, CsCl-purified Privateer particles were separated by SDS-PAGE on a Bio-Rad 4-20% Tris-glycine gel with ~1011 pfu/mL per lane. The gel was stained with Coomassie blue (0.1% Coomassie R-250, 10% acetic acid, 40% methanol). Prestained SeeBlue-Plus2™ (Invitrogen, Carlsbad, CA, USA) was used as a molecular weight marker. This was for visual inspection only, as whole phage particles, rather than individual bands, were the starting sample for mass spectrometry.

A sample of whole Privateer particles from CsCl-purified stock was prepared for mass spectrometry analysis by methanol-chloroform precipitation with a 6:1.5:2 methanol:chloroform:water ratio. Approximately 50 µg of extracted phage particles were treated with 5% SDS, then a nuclease cocktail that contained DNase at 10 µg/mL (Type II; Sigma-Aldrich, St. Louis, MO, USA) and RNase at 50 µg/mL (Type IIIa; Sigma-Aldrich, St. Louis, MO, USA) in 10 mM Tris at pH 7.0, for 15 min on ice. The treated sample was then applied to an S-Trap column (Protifi, Melville, NY, USA) and processed according to the manufacturer protocol. All proteins were reduced on-column with DTT and alkylated with iodoacetamide prior to digestion with modified porcine trypsin (Promega, Madison, WI, USA). Resulting peptides were collected and dried prior to LC-MS/MS analysis.

After shipment, the sample was resuspended in 0.5% acetic acid/0.005% trifluoroacetic acid for HPLC-electrospray ionization-tandem mass spectrometry (HPLC-ESI-MS/MS) accomplished on a Thermo Fisher LTQ Orbitrap Velos Pro mass spectrometer (Thomas et al., 2010; Weintraub et al., 2018). Mascot (Matrix Science; London, UK) was used to search the MS files against a database of sequences for predicted proteins for phage Privateer (146 sequences; 28,814 residues) and the SwissProt database (SwissProt 2019_10 (561,356 sequences; 201,858,328 residues)) similar to previously described studies of other phage proteins (Thomas et al., 2010; Weintraub et al., 2018). Subset searching of the Mascot output by X! Tandem, determination of probabilities of peptide assignments and protein identifications, and cross correlation of the Mascot and X! Tandem identifications were accomplished by Scaffold 4 (Proteome Software).

Phylogenetic analysis

Homologous structural proteins for Privateer portal protein (NCBI accession QIN94795.1), scaffolding protein (QIN94797.1), major capsid protein (QIN94798.1), and tail tubular protein (QIN94802.1) were identified via CoreGenes3.5 (Turner et al., 2013). Alignments were generated with MUSCLE (Edgar, 2004). Maximum likelihood analysis was run with a bootstrap value of 100 via Phylogeny.fr, and the tree was constructed using TreeDyn (Castresana, 2000; Guindon & Gascuel, 2003; Anisimova & Gascuel, 2006; Chevenet et al., 2006; Dereeper et al., 2008, 2010).

Results

Privateer isolation and characterization

Bacteriophage Privateer was isolated in 2019 from activated sludge obtained at a wastewater treatment facility in Navasota, Texas. Plaques produced by Privateer when plated on P. mirabilis strain ATCC 35659 are clear and approximately 0.15-mm in diameter (Fig. 1A). The phage was purified via ultracentrifugation on a cesium gradient for use in further proteomic and infection cycle characterization (Fig. 1B). Transmission electron microscopy (TEM) imaging of the negatively-stained purified phage particles demonstrated that Privateer is a podophage with C3 morphology, typified by the elongated capsid (Fig. 1C) (Ackermann, 2001). The virion head is ~140-nm in length and ~35-nm in diameter, with a short ~10-nm tail.

Figure 1 Privateer isolation and imaging.

(A) Privateer clear plaques with a diameter of ~0.15-mm on P. mirabilis ATCC 35659 lawn. (B) The blue band containing the purified Privateer particles following CsCl step-gradient ultracentrifugation. (C) Privateer transmission electron micrograph displaying virions ~145 × 35 nm, with a stubby ~10-nm tail.

Privateer infection kinetics and lysis

To assay Privateer growth, we determined that nearly 80% of the phages were adsorbed to host cells within eight minutes post-phage addition (Fig. 2A). From a one-step growth curve, an average 138 ± 56 burst size was calculated (Fig. 2B). At higher multiplicity of infection (MOI), Privateer induces host growth cessation within 40 min, but overt lysis does not begin until 100 min and lasts ~40 min in bulk culture (Fig. 2C). Individual infected host cells were observed by oil immersion to reveal an abrupt and destructive explosion associated with cell lysis after comparatively little change to the cell appearance through that point (see Movies 1, 2 and 3).

Figure 2 Characterization of Privateer infection.

(A) Privateer adsorption curve, where phages not adsorbed to host cells were detected relative to input. P/Po = free phages at time point/free phages at 0 min. (B) Privateer one-step growth curve, where plaque-forming units were quantified after MOI = 0.01 infection. (C) Lysis curve at MOI = 10, where OD550 of host liquid culture was measured over time. Representative instances of three replicates are shown.

Phage host range

The host range for phage Privateer was assayed via spot-dilution against a bacterial panel including six P. mirabilis strains, mainly uncharacterized clinical isolates (Table 1). Privateer produced smaller plaques on four P. mirabilis strains at a significantly reduced efficiency of plating (EOP <10−3). Additionally, the phage produced faint clearing, distinctive of killing from without, on P. mirabilis ATCC 43071 in high titer spots (>109 pfu/mL). Privateer did not plaque on strain B446 or any other tested Gammaproteobacteria: Escherichia coli MG1655, Klebsiella pneumoniae 1776c, and Serratia marcescens D1.

Privateer genome annotation

The 90.71-kb Privateer genome was assembled into a single contig with 636.7x coverage. The Privateer genome encodes 144 proteins, at an 88.3%, coding density, of which 43 were assigned predicted functions (Fig. 3). A 34.52% G+C content for the Privateer genome stands in contrast to the consistent 39% G+C content of Proteus genomes (Falkow, Ryman & Washington, 1962; Pearson et al., 2008; Sullivan et al., 2013).

Figure 3 Privateer genome plot.

The predicted genes are color-coded corresponding to the functional categories of their protein products. The label for virion-associated proteins detected in purified phage particles by mass spectrometry are bordered with red.

Comparative genomics

Phage Privateer shares 83.1% nucleotide sequence identity with Cronobacter phage vB_CsaP_009. Privateer and vB_CsaP_009 both share low nucleotide sequence identity (<10%) with all other known phages. Despite low nucleotide similarity, BLASTp results demonstrated that in addition to 128 similar proteins between Privateer and vB_CsaP_009, several other phages also encode numerous similar proteins. These include Cronobacter phage vB_CsaP_GAP52 (71 proteins), Salmonella phage 7-11 (65 proteins) (Kropinski, Lingohr & Ackermann, 2011), Aeromonas phage Lah_6 (45 proteins) (Kabwe et al., 2020) and Vibrio phage Vp_R1 (34 proteins) (Ren et al., 2019). TEM imaging demonstrated that phages 7-11, Lah_6, and Vp_R1 each belong to the C3 podophage morphotype. Privateer also shares similar proteins with phages in the Kuravirus genus, including Escherichia phages phiEco32 (33 similar proteins) (Savalia et al., 2008), Paul (33 proteins) (Holt et al., 2019) and vB_EcoP_SU10 (32 similar proteins) (Mirzaei et al., 2014). The Kuravirus genus is within the Podoviridae family and consists of C3 morphotype (prolate) phages, with phiEco32 as the exemplar. We conclude that Privateer is phiEco32-like, and below compare and contrast with a select group of C3 morphotype phages (Table 2; Fig. 4).

Figure 4 Comparative genomics among similar phiEco32-like phages.

Genome organization and comparison of protein identities for Proteus phage Privateer with selected C3 morphotype podophages. Cronobacter phage vB_CsaP_009 and Cronobacter phage vB_CsaP_GAP52 have the highest nucleotide identity with Privateer, and are all unclassified within the Podoviridae family. Escherichia phage Paul, Escherichia phage vB_EcoP_SU10, and Escherichia virus phiEco32 are phages in the Kuravirus genus. Proteins sharing significant sequence identity with at least 10−20 BLASTp expectation value are linked via gray bands (see “Materials and Methods”).

Virion protein analysis

SDS-PAGE and mass spectrometry analyses of Privateer particles verified the presence of ten predicted virion-associated proteins, including the portal protein (QIN94795), minor (QIN94799) and major (QIN94798) capsid proteins, and tail tubular protein (QIN94802) (Fig. 5). The major capsid gene contains a programmed ribosomal frameshift signal (GGGAAAG) predicted to result in a second, larger, minor capsid protein, as demonstrated for phage T7 proteins 10A and 10B, among others (Sipley et al., 1991). The two bands for capsid protein products expected from a translational frameshift were observed by SDS-PAGE (Fig. 5A). Both suspected capsid protein bands are migrating at an apparent molecular mass higher than the calculated molecular weight. While this could be an artifact of comparison to a stained protein ladded, further characterization is required to verify the state of the capsid protein. The minor capsid protein was detected in virions via mass spectrometry (Fig. 5B). Although, the use of whole virions containing a mixture of all structural proteins precludes a definitive identification of the major capsid protein, the capsid and its frameshifted product were observed in both phiEco32 and vB_EcoP_SU10 (Savalia et al., 2008; Mirzaei et al., 2014), indicating its conservation among phiEco32-like phages.

Figure 5 Analysis of Privateer virion proteins.

Proteins of phage Privateer identified by SDS-PAGE and mass spectrometry. (A) CsCl purified phage particles were separated on a 4–20% Tris-glycine SDS-PAGE gel. Molecular masses (in kiloDalton) of the protein ladder are displayed to the left of the gel. The white and black arrowheads indicate the expected location for minor and major capsid bands, respectively. (B) Table of mass spectrometry results for trypsin-digested Privateer proteins from whole phage particles. The total spectrum count is equal to the total number of total peptide spectral matches assigned to the protein, and the unique peptide count is equal to the number of peptide sequences exclusive to the protein. (C) Aligned sequences of the annotated major and minor capsid proteins. The highlighted regions indicate peptides identified via mass spectrometry.

Structural region phylogenetic comparison

A phylogenetic tree depicting the evolutionary relatedness of Privateer and selected C3 podophages on the basis of the presence of major capsid and three other conserved structural proteins suggests that the C3 phages descend from a common ancestor and diverge into two distinct clades: the Kuravirus and non-Kuravirus phages (Fig. 6). The distances between the non-Kuravirus phages are larger than those between the tightly grouped (e.g., more closely related) Kuravirus phages.

Figure 6 Phylogenetic tree based on phage morphogenesis proteins.

The phylogenetic tree was built using four conserved structural proteins: portal, scaffold, major capsid and tail tubular protein. Branch support values are displayed in red. See “Materials and Methods” for additional details and accessions.

Discussion

Though P. mirabilis is a significant human pathogen resulting in UTIs and CAUTIs, Privateer is among a small number of characterized virulent phages for this host. The prolate morphology observed in Privateer virions classifies it among C3 morphotype phages, with its closest related phages in the sequence database also primarily infecting Proteobacteria hosts. The limited host range data presented here should be further investigated in comparison to other P. mirabilis phages, and likely in close conjunction with related phages of other Enterobacterales. Genome inspection, and comparison with other phages, reveal that the packaging and structural genes are organized in a modular fashion, however, the remainder of the Privateer genome is less distinctly grouped by gene function. Below, we discuss the genetic repertoire for both expected functional predictions and several features unusual to phage genomes.

DNA packaging

The DNA packaging protein terminase (NCBI accession QIN94794), is highly conserved across C3 morphotype phages examined in this report. The terminase shares ~60% amino acid sequence identity with Kuravirus phages and ≥70% identity with the non-Kuravirus phage terminases. The packaging strategy for the Kuravirus phages phiEco32, Paul, and vB_EcoP_SU10 was predicted to involve direct terminal repeats (<200-bp) similar to the packaging approach of Escherichia phage T7 (Chung & Hinkle, 1990). Using the PhageTerm algorithm, Privateer is predicted to have novel end types, and the non-Kuravirus phages discussed here have not been investigated in this way.

Lysis

Three lysis proteins were identified: an inner- and outer-spanin (QIN94818, QIN94817) and an endolysin endopeptidase (QIN94840). The two spanin proteins are located immediately following the last structural protein (QIN94816), while the endolysin is situated ~12-kb downstream (Fig. 3). The endolysin protein has 96% amino acid identity to its homolog in vB_CsaP_009, and 52% similarity to vB_CsaP_GAP52 endolysin. The shared conserved domains with high amino acid identity suggest that these endolysin proteins might be active against the peptidoglycan of both bacterial cells. A holin gene could not be reliably identified in the Privateer genome. Ten proteins with unknown functions contain at least one transmembrane domain, and are therefore potential holin candidates (Young, 2002). Unlike Privateer, the Kuravirus phages each encode a putative holin and endolysin among the morphogenesis genes, possibly a grouping of late genes (Fig. 4).

Biosynthesis

Privateer encodes several biosynthesis proteins, including four ribonucleotide reductases (QIN94826, QIN94860, QIN94873, QIN94874), two nicotinate ribosyltransferases (QIN94828, QIN94907), a deoxycytidylate deaminase (QIN94864), a deoxyribosyltransferase superfamily protein (QIN94829), and a polynucleotide kinase (QIN94912). The only two Privateer biosynthesis proteins with homologs in Kuravirus phages are the deoxycytidylate deaminase and a glutamine amidotransferase domain-containing protein. Given the several nucleotide metabolism proteins present in Privateer, but not in Kuravirus genomes, efficient replication of Privateer DNA may require additional machinery.

Stress response and regulation

The genome additionally encodes four proteins associated with responses to chemical stress: a thioredoxin (QIN94863) and three tellurium resistance, or ter, proteins (QIN94906, QIN94916, QIN94918). When expressed as elements of bacterial genomes, ter proteins are involved in cellular processes such as the regulation of chemical stress responses and phage resistance (Whelan, Colleran & Taylor, 1995; Anantharaman, Iyer & Aravind, 2012). Tellurite resistance has also been positively correlated with phage resistance (Tomás et al., 1984). The specific role of ter proteins expressed by phage genomes has not yet been established experimentally. Although the three ter stress response proteins are absent in the Kuravirus-like phages, the thioredoxin is conserved across each C3 morphotype phage examined in this report. Thioredoxins facilitate a wide range of functions in different organisms, including the reduction of potentially harmful reactive oxygen species and controlling the structure or catalytic activity of proteins via manipulation of thiol-group redox states (Arnér & Holmgren, 2000). Additionally, thioredoxins play critical roles in nucleotide metabolism as electron donors for ribonucleotide reductases and other biosynthesis proteins. Past structural and enzymatic assays demonstrated that Escherichia phage T7 DNA polymerase binding affinity for primer sequences is improved ~80-fold by coupling with host thioredoxin (Huber, Tabor & Richardson, 1987; Bedford, Tabor & Richardson, 1997). HHpred analysis predicted significant structural similarity between Privateer DNA polymerase (QIN94848) and T7 DNA polymerase (NP_041960). Privateer thioredoxin (QIN94863) is also predicted to have significant structural similarity at >99% probability across >75% of the protein to thioredoxins of various organisms by HHPred. We hypothesize that the Privateer thioredoxin may similarly couple with its DNA polymerase to improve processivity, thus enhancing the rate of phage DNA replication, and explaining the presence of homologs for both proteins in each C3 phage examined in this study.

DNA replication and recombination

Privateer encodes proteins involved in DNA replication and recombination including a DNA adenine methylase (QIN94821), DNA polymerase (QIN94848), DNA primase/helicase (QIN94866), DNA ligase (QIN94832) and two exonucleases (QIN94834, QIN94865). The Kuravirus genomes encode homologs for the DNA polymerase, DNA primase/helicase, and both exonucleases. A vB_CsaP_009 protein (BBU72708) shares 92.82% amino acid sequence identity with the DNA adenine methylase, yet no homologs nor additional DNA methylases were found within the other C3 phages (including Cronobacter phage vB_CsaP_GAP52). This could indicate that Privateer DNA within P. mirabilis requires methylation for either increased transcription efficiency or to defend against a host anti-phage response not active in another host species (Murphy et al., 2013).

Privateer additionally encodes an RtcB-like RNA ligase (QIN94896), which has a homolog in vB_CsaP_009 and vB_CsaP_GAP52. Studies have shown RtcB RNA ligase involvement in ribosome homeostasis and RNA repair in bacteria and certain eukaryotes (Tanaka, Meineke & Shuman, 2011; Chakravarty et al., 2012; Engl et al., 2016). The Privateer RtcB-like RNA ligase is located ~1-kb downstream from a cluster of three tRNA genes (Arg anticodons TCT and CCT, Met/Ile2 anticodon CAT) (Chan & Lowe, 2019), and the genome also encodes a fourth tRNA gene (Trp anticodon CCA) ~15 kb further downstream (Fig. 4). The Kuravirus phages code for one tRNA gene at most, though not the same tRNA. Privateer’s RtcB-like RNA ligase may ensure tRNAs remain stable and in sufficient quantity within the host to facilitate the rapid production of phage proteins, as suggested for certain mycophages encoding >20 tRNAs (Pope et al., 2014). Phages often encode transfer RNA genes if their specific host does not provide adequate tRNAs for efficient phage protein synthesis, or as a strategy to allow host switching (Bailly-Bechet, Vergassola & Rocha, 2007; Delesalle et al., 2016). There is not a strong correlation between codon usage in the host and Privateer (data not shown). Therefore, the lack of tRNA conservation within Proteus phages does not shed light on a specific host deficiency.

Additional genome features

The only Privateer protein with an explicit role in transcriptional regulation was an alternative RNA polymerase extracytoplasmic function (ECF) sigma factor (QIN94836). Homologs for the ECF sigma factor are present in the Kuravirus phages and the phiEco32 RNA polymerase sigma factor was experimentally shown to bind to E. coli RNA polymerase (Savalia et al., 2008). An autoregulation sequence within the deduced promoter binding region near the -35 sequence has weak conservation of an expected homopolymeric T-tract, but other elements vary (Mascher, 2013; Guzina & Djordjevic, 2017). The phage ECF sigma factor subgroup is considered an outlier compared to canonical families, and may employ flexible binding strategies that allow promoter recognition (Guzina & Djordjevic, 2016). Additionally, no clear candidates for a soluble or membrane-bound anti-sigma factor were annotated in the Privateer genome.

The predicted DNA-binding Dps family protein (QIN94861) was detected by mass spectrometry with purified virions (Fig. 5). While its function in Privateer infection has not been verified, Dps family homologs are also present in the Kuravirus phages.

Conclusions

Privateer is the first prolate podophage reported to infect Proteus hosts, and exhibits typical characteristics for Caudovirales infection, including explosive lysis behavior. The host range here assayed for Privateer is relatively narrow, and its closest known genetic relative infects Cronobacter, another Proteobacteria. In the Privateer genome we predicted protein functions for morphogenesis, lysis, DNA replication and recombination, and biosynthesis among others, though these displayed limited modular organization. Proteomic analysis on whole phage particles verified the presence of predicted structural proteins, including a longer capsid protein hypothesized to arise by a translational frameshift mechanism. Phylogenetic analysis performed using the amino acid sequences of four conserved structural proteins (including the major capsid protein) illustrated Privateer relatedness to other C3 morphotype bacteriophages. Some Privateer proteins share amino acid identity with proteins found in other C3 phages, including the phages classified in the Kuravirus genus. Several additional interesting genome features were noted, including the presence of an RtcB-like RNA ligase, an ECF RNA polymerase sigma factor, and three ter stress response proteins. Individual phage characterizations such as the one presented here are an essential beginning step for building a larger panel of diverse Proteus phages to investigate within infection models in combination. As less than 30 Proteus phage genomes have thus far been sequenced, a broader collection of Proteus phages should be sequenced and studied at the bench if phages like Privateer are to be used as alternative treatments of CAUTIs in the future. Current comparisons to related phages isolated on other bacterial hosts can promote such studies in both therapeutic and basic phage research contexts.

Supplemental Information

Supplemental Information 1 Raw data for Figure 2.

Click here for additional data file.

Supplemental Information 2 Raw data for Figure 5.

Original acquisition of SDS-PAGE shown in Fig. 5A.

Click here for additional data file.

Supplemental Information 3 Privateer lyses infected P. mirabilis host cell.

An individual P. mirabilis cell infected with Privateer at MOI = 3 infection observed between 60 and 80 min post-infection at 100X magnification. Images for this individual cell were recorded through complete lysis, which occurs at ~30 s in this 42-s time-series video.

Click here for additional data file.

Supplemental Information 4 Privateer lyses infected P. mirabilis host cell.

A second individual P. mirabilis cell infected with Privateer at MOI = 3 infection observed between 60-80 minutes post-infection at 100X magnification. Images for this individual cell were recorded through complete lysis, which occurs at ~9 s in this 20-s time-series video.

Click here for additional data file.

Supplemental Information 5 Privateer lyses infected P. mirabilis host cell.

A third individual P. mirabilis cell infected with Privateer at MOI = 3 infection observed between 60-80 minutes post-infection at 100X magnification. Images for this individual cell were recorded through complete lysis, which begins at ~26 s into this 34-s time-series video.

Click here for additional data file.

The initial stages of this project were completed in partial fulfillment of BICH464 Bacteriophage Genomics requirements, an undergraduate course at Texas A&M University in the Department of Biochemistry & Biophysics. We are grateful for the advice and assistance from the Center for Phage Technology staff, in particular Mei Liu, James Clark, and Isla Hernandez.

We acknowledge and thank the Protein Chemistry Laboratory at Texas A&M University for their guidance and assistance in preparing sample for the mass spectrometry analysis. Mass spectrometry analyses were conducted in the Institutional Mass Spectrometry Laboratory of the University of Texas Health Science Center at San Antonio. The expert technical assistance of Sammy Pardo, Dana Molleur, and Susan T. Weintraub is greatly appreciated.

Additional Information and Declarations

Competing Interests

Author Contributions

DNA Deposition

Data Availability

New Species Registration

The authors declare that they have no competing interests.

James E Corban conceived and designed the experiments, performed the experiments, analyzed the data, prepared figures and/or tables, authored or reviewed drafts of the paper, and approved the final draft.

Jolene Ramsey conceived and designed the experiments, performed the experiments, analyzed the data, prepared figures and/or tables, authored or reviewed drafts of the paper, and approved the final draft.

The following information was supplied regarding the deposition of DNA sequences:

Genome and associated data for phage Privateer are available at GenBank: BioProject PRJNA222858, accession: MT028297, BioSample: SAMN14002513. Additional data is available at the Sequence Read Archive accession no. SRR11024936.

The following information was supplied regarding data availability:

The mass spectrometry raw data has been deposited at MassIVE and is available at DOI 10.25345/C5DZ18.

The following information was supplied regarding the registration of a newly described species:

This proposed new species, Proteus phage Privateer, has been submitted to the International Committtee on the Taxonomy of the Viruses (ICTV) for consideration. This name is only valid and official after the ICTV has approved the name, and it has been ratified by the membership here: https://talk.ictvonline.org/files/proposals/taxonomy_proposals_prokaryote1/m/bact01/10159 and here: https://talk.ictvonline.org/files/proposals/taxonomy_proposals_prokaryote1/m/bact01/10160.

The NCBI has provisionally assigned a taxid (NCBI:txid2712958) for Proteus phage Privateer

https://www.ncbi.nlm.nih.gov/Taxonomy/Browser/wwwtax.cgi?id=2712958.

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
