# Peer review of "Characterization and complete genome sequence of Privateer, a highly prolate Proteus mirabilis podophage"

_PeerJ, doi:10.7717/peerj.10645_

## Round 0.1 · original submission · Minor Revisions

All three reviewers were supportive of the manuscript's publication and suggested minor revisions for improving the final version.

·

Basic reporting

Basic Reporting
The submission is a comprehensive, well-written report of the characterization of a new podophage with potential clinical applications. It originates from a research group with extensive experience in bacteriophage characterization. The article structure is appropriate and the methods well-described. The introduction spends time discussing a group of articles using polyvalent phage preparations with insufficient discussion of the host being studies and the biological importance of phage in the gut, their role in the microbiome and the sewage source. The discussion is similarly rather sparse.

Experimental design

Experimental Design
The experimental design is appropriate and comprehensive, characterizing the phage titer, infectivity characteristics, genome features and proteins amenable to mass spectrometry. The methods are presented in admirable detail (with a few minor suggestions below). The methods appear to be well-established methods of phage analysis.

Validity of the findings

Validity of the Findings
The findings appear credible and internally consistent. The figures are well-presented and mostly well-done.

Additional comments

Specific Comments with line numbers



Overall admirably detailed methods
86 Phage Storage conditions?
80 nuclease supplier?
104 Typo: Hernanded
140-PhageTerm in Galaxy? Or BASH?
148-Cutoff? Rho-independent termination scan not mentioned again….were none found?
The Genbank has record noting that frameshift was verified by MS. Nice!
Both GENBank and ShortReadArchive records appear complete
173 Morphology also reminiscent of kuraviruses
179 Appreciate that the software for the figures were acknowledged. A lot of people ignore that intellectual property.
197 Dried sample storage? Was it reconstituted in methanol or……….?
200 The MS analysis should be cited. There are a number of papers on spectral counting statistics for peptide identification would be appropriate since there is some correlation implied between the SDS-PAGE gel band density and MS spectral count quantification. Is the MASCOT procedure proprietary? Also, explanation of the distinctions between the total spectrum count and unique peptide count would be a good addition.
221 infective characterization? Odd phrase
222- Delete citation; it’s in the methods
227 –“reproduction” replace with ‘infection kinetics’
238 Any discussion of P. mirabilis B446? It is strikingly different. Any characterization of P. mirabilis B446 that one could speculate about differences? Host range is not mentioned in the discussion.
299 “Assigned predicted functions are here discussed for both the
300 expected genetic repertoire, as well as several unusual genome features”. I understand what is being said here but even in passive voice, that can be better expressed.
307 You could search the non-Kuraviruses for DTRs, yes? Was that done?
315 location of Kuravirus holin and endolysin?
331 “The specific role of ter proteins expressed by phage
331 genomes has not yet been explored” Well, it has been explored but not established.
341 How much is the similarity between Privateer thioredoxin and T7? Or is this just raw speculation?





Fig 2B nice looking curve; burst size of about 100 is as to be expected
Would prefer reporting of the three values not a representative curve for 2A and 2B
Can’t find this reference anywhere. Is it “In Press” or something? If status is just “accepted”, that should be noted.
Ramsey J, Rasche H, Maughmer C, Criscione A, Mijalis E, Liu M, Hu JC, Young R, Gill JJ. 2020. Galaxy and Apollo as a biologist-friendly interface for high-quality cooperative phage genome annotation. PLOS Computational Biology.

Overall, very solid paper. My main criticism is that there is insufficient discussion of the host range, yet an attempt to extrapolate the host range to Proteus spp. and further extrapolate that to clinical applications. All justifiable, but a few lines to delineate those ties are needed.
The use of Privateer against CAUTIs is mentioned in the introduction and conclusion, but it seems pro forma for funding purposes. There is room for more discussion that would justify that connection. Although the host range doesn’t provide strong evidence that the isolated host is close to the primary host of the phage, the infection and burst data clearly demonstrate that it is capable as acting as an anti-bacterial for relatives of P. mirabilis. The Milo, Ujmajuridze and Maszewska papers cited use polyvalent phage preparations, so there is some advantage to characterizing individual phages. Discussion of the host range of Cronobacter phage vB_CsaP_GAP52, in particular should be discussed. What is the level of similarity between endolysin of those two phages? (Holin in Cronobacter phage vB_CsaP_GAP52 has not been identified, either).

Reviewer 2 ·

Basic reporting

Very nicely written.

Experimental design

The research plan was well described, and methods were easy to follow.

Validity of the findings

Overall the data and conclusions were valid and well described. A few comments are listed to be addressed to clarify a few of the results and figures.

Additional comments

Specific Comments for the authors to address:
line 51: A space is needed between lacking and (

line 179 : Citation for software that has no manuscript typically has a link to download. This is not listed in the methods or the reference list for Inkscape.

line 121-122: Please add reference to table 1 with the host range strains of interest.

line 228-235: Figure 2 and the methods indicates three replicates of adsorption curves, one-step growth curves, and lysis curves. In the results, figure, and raw data for figure 2 only the representative image is shown. Please provide statistical analysis for each of these metrics. If necessary please put the other replicates data in supplemental material.

line 259-261: Many of the phages listed in the text which have protein similarity to Privateer are not shown in the comparative genomics figure 4. Please update text to reflect those that are described in figure 4 versus those that are only described in the phylogenetic tree in figure 6.

In the description of comparative genomic analysis (lines 258-265) with other phages proteins, which types protein are conserved (structural or throughout genome)?

Figure 1A: If possible increase resolution on this image.

Figure 4: Please indicate the species of each virus in the legend since these phages come from a diverse group of bacteria. Does the grey represent a certain percentage homology between each protein? If so can you indicate what percent homology that is in the legend.

Figure 5: Can you add rational for the shift to a higher molecular mass for the major and minor capsid band? The molecular mass listed in the table is 35kD and 53kD but appears to be closer to 38kD and 60kD in the gel.

Table 1: Please add EOP data for each strain, some this data is mentioned in the text on line 240 but is not replicated in the table.

Reviewer 3 ·

Basic reporting

In this article, Corban et al. isolated and characterized a novel prolate podophage, Privateer, that infects a relatively narrow range of Proteus mirabilis strains. The article very well written, detailed, and easy to follow. I am extremely impressed by the meticulous discussion on putative phage encoded proteins. For example, instead of just reporting that Privateer encodes a potential thioredoxin, the authors explained the role of thioredoxin in bacterial stress response and how it has been shown to influence the primer binding affinity of phage T7 DNA polymerase. I have some minor comments/suggestions for the authors:

Line 50: Introduce space between “lacking(”
Line 145: I could not find Ramsey et al., 2020 and the reference does not have a DOI. Is the article in press? I could not find a BioRXiv version of this reference either. This will be a very useful reference for the entire phage community.
Line 228: From Fig.2A, seems like it should be “nearly 80%” instead of “80% of the phages were adsorbed to host’.
Line 307: Can the authors expand on “Privateer is predicted to have novel end types”?

Experimental design

Line 69-72: Please provide a detailed protocol for the phage isolation technique here.

Validity of the findings

No comment

Additional comments

No comment

---

## Round 0.2 · accepted · Accept

Thank you for thoroughly addressing all the reviewer's comments!